

# RARE: right algorithm for the right errand; a multi-model machine learning-based approach for tourism routes and spots recommendation

Ling Luo

Henan Vocational College of Information and Statistics, Zhengzhou, Henan, China

## ABSTRACT

With the globalization of the economy, tourism has emerged as a significant sector of entertainment and economic growth. Optimizing tourist attractions and routes has become crucial in modern travel planning, driven by the increasing demand for personalized recommendations. However, traditional static route-based algorithms struggle to adapt to the rapid expansion of the tourism industry, necessitating the development of dynamic, machine-learning-driven solutions. This study introduces a novel tourism recommendation system integrating multiple machine learning algorithms to provide personalized tourist spot and route recommendations. The proposed approach models the tourist map as a 2D grid of interconnected nodes, allowing for dynamic and adaptive recommendations. The framework employs long short-term memory (LSTM) for spot relevance prediction, support vector machine (SVM) for spot name classification, and depth first search (DFS) for optimal route generation. A k-means clustering approach is also utilized to designate a cluster leader (CL) responsible for managing node information within a specific zone. By inputting a simple textual query, tourists receive optimized travel routes tailored to their preferences, incorporating relevant attractions. The model is implemented in a Python-based environment and evaluated using an augmented Travel Recommendation dataset from Kaggle. Experimental results demonstrate the model's effectiveness in enhancing tourism planning and user experience, showcasing its potential for advancing intelligent tourism solutions.

## INTRODUCTION

Tourism is not a mere hobby; it is a form of entertainment that enhances the quality of people's lives (*Lyu & Han, 2022*). Tourists prefer to visit scenic and historical spots for serenity, solace, non-formal education, revolutionary feelings and adventurous experiences. Whether it is honeymoon tours, island tours, or red tourism, people prefer to enjoy their leisure fully, full of natural calmness and free from worldly worries. However, to endure a calm and congenial trip, route selection based on personal priorities is of utmost importance (*Flisberg et al., 2012*). Traditionally, tourists relied on guidebooks,

Corresponding author
Ling Luo, Ling__Luo@163.com

magazines, and static websites for travel recommendations. These methods lag far behind in addressing personalized priorities and dynamic demands of modern-day tourism. In the current day and age, machine learning (ML) technology may be used to revolutionize tourism, mainly to select optimal travel routes.

With the advent of internet technology, easy availability of information has become the norm of the day. This unprecedented advancement has made a remarkable surge in the tourism industry. The progress in tourism is further fueled by the emerging AI methods, which make tourism-related data ubiquitous and ensure personalized route planning (*García-Madurga & Grilló-Méndez, 2023*). Consequently, nowadays, tourists are privileged with many options in terms of route selection, availability of facilities and other prudent options. However, besides various challenges, man's search for an effortless model to exploit personal preference, budgetary consideration, and time constraints has yet to be fulfilled (*García-Madurga & Grilló-Méndez, 2023*).

Building on the success of machine learning (ML) in education, healthcare, and sports, advanced AI models can revolutionize the tourism industry by providing personalized travel recommendations. This research extends the role of AI in tourism by integrating long short-term memory (LSTM), support vector machine (SVM), and k-means clustering to generate travel routes based on tourists' preferences for attractions. The tourist map is modeled as a 2D grid of interconnected nodes, utilizing mental map (MM) and distance vector (DV) structures of Dijkstra's presentation. Since the dataset lacks distance information, additional distance and direction columns are incorporated before training. Dijkstra's algorithm is employed to calculate distances between nodes. A k-means clustering approach selects a cluster leader (CL) responsible for managing attraction data within a given zone, with CLs dynamically sharing updates.

At the CL level, an LSTM network is trained on dominant attraction data, while an SVM classifier predicts spot names from spot features. Given a simple spot attraction query, LSTM generates attraction-related sequences, which are then processed by SVM to predict the optimal target spot name. The depth first search (DFS) algorithm then traces possible routes from the source to the target destination.

The model is implemented in Python using an augmented Travel Recommend Kaggle's dataset from Kaggle and evaluated on 52 randomly selected tourism queries, achieving a promising accuracy of 86%. These results validate the effectiveness and applicability of the proposed approach in enhancing personalized travel planning. The schematic representation of the RARE method is shown in Fig. 1.

Rest of the article is structured into five sections. A literature review is covered in 'Literature Review'. 'Methodology' is about the detailed methodology, whereas implementation details are presented in 'Implementation Details'. Experimentation and evaluation are reported in 'Experimentation and Result Analysis'. The last section, the 'Conclusion', concludes the article with future research directions.

## LITERATURE REVIEW

With the easy availability of transportation facilities, tourism has grown to the level of a beneficial industry. The industry plays a significant role in the economic growth of the

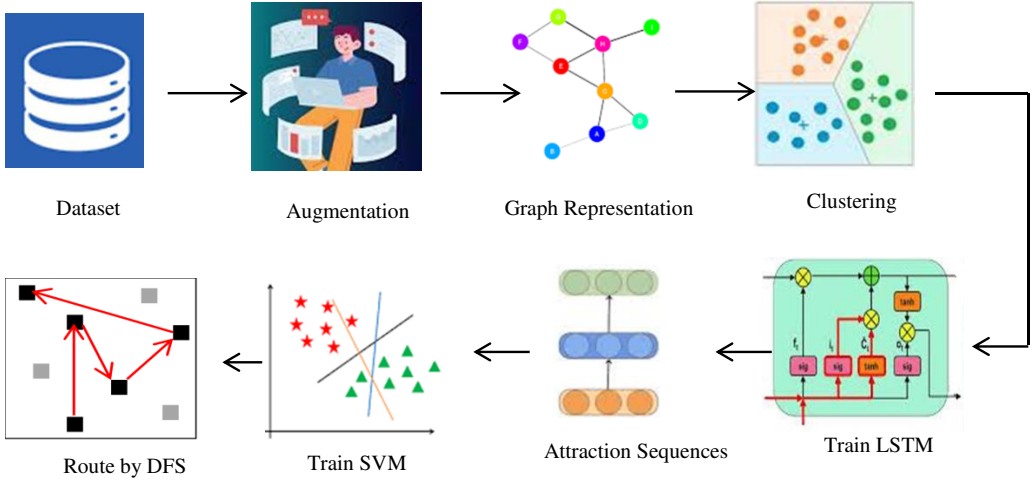

**Figure 1 Schematics of the RARE method.**

offering country (*Malik & Kim, 2019*). The effortless accessibility of tourist attraction information and luxuries enhances tourism (*Basiri et al., 2018*). The location attainment technologies and social and cellular networks have contributed to the uplifting of the tourism industry (*Zhu et al., 2019*; *Cao et al., 2021*). However, as mentioned in *Basiri et al. (2018)*, challenges in tourism are yet to be resolved. The key challenges include how to tour, what to tour, and where to tour, specifically whence the destination is unfamiliar (*Du et al., 2019*). The prominent issue amthemu'. A number of research works have been proposed to cater to 'how to tou'. Besides the traditional approaches, data mining techniques have been utilized to help tourists 'how to tou' (*Yuan et al., 2016*).

Due to individual differences, tourists' preferences for various tourist attractions are different (*Bao et al., 2015*; *Hang et al., 2018*). Keeping in view the tourist preference, route recommendations can be categorized into destination recommendations and route recommendations. Where the former focuses on a single attraction satisfying the tourists' interest, the latter is about suggesting multiple routes that suit a tourist's interest. In either case, suggesting suitable routes for multiple attractions is challenging (*Pu, Chen & Hu, 2012*). Most of the conventional path-planning recommendation approaches are based on the shortest path (*Pu, Chen & Hu, 2012*). A number of methods have been suggested in this scenario, including the A* approaches (*Singh et al., 2018*; *Song, Liu & Bucknall, 2019*), particle swarm optimization (PSO) (*Pongchairerks & Kachitvichyanukul, 2016*), and Genetic Algorithm (GA) (*Ghosn, Drouby & Harmanani, 2016*), ACO algorithm (*Mirjalili, Dong & Lewis, 2020*), simulated annealing (SA) (*Demiral & Işik, 2020*), and graph-based algorithm (*Skinderowicz, 2022*). The vehicle routing planning problem (*Bell & McMullen, 2004*) and traveling salesman problems (*Zou, Yang & Zhao, 2024*) are also pertinent to mention in this regard.

Recommendation systems based on user textual data have been suggested to facilitate tourists (*Du et al., 2019*). Similarly, the system of *Zhang et al. (2012)* works on travel notes to help tourists, whereas that of *Chen et al. (2025)* uses tourist records. The research work

of *Ye et al. (2011)* is about improving classification preference by using textual and geographical information. The information cumulative approach of *Qiao et al. (2024a)* extracts data from tourism-related blogs to easily identify attraction of interest. The work of *Yuan et al. (2016)* utilizes the big data mining approach to trace the most popular spot in a region. The Photo2Trip method proposed in *Lu et al. (2010)* exploits photos that are geotagged adequately for recommending travel routes. The pattern-ware system of *Wei, Peng & Lee (2013)* searches out the top K paths to scenic spots and recommends the optimal one. The framework of *Hu et al. (2015)* recommends urban attractions using the DBSCAN clustering algorithm. The K+V-DBSCAN algorithm in *Pla-Sacristán et al. (2019)* is suggested to accurately identify a particular zone's most significant tourist attractions. The context-aware system of *Majid et al. (2013)* exploits geotagged graphical data to recommend places based on the interest of tourists. Similarly, the *point-of-interest*-based system of *Zuo et al. (2024)* utilizes sentimental attributes for attraction recommendation. Tourists prefer to use personalized recommendation systems (*Santos et al., 2019*). Therefore, collaborative techniques have also been utilized for effective attraction recommendations (*Bin et al., 2019*). Such systems give tourists various friendly options to improve recommendations (*Park, Park & Hu, 2021*). A fuzzy ontology method is proposed in *Abbasi-Moud et al. (2022)*. The method analyzes the tourists' reviews for future consideration. The model of *Hong & Jung (2021)* analyzed the reviews and rated significant attractions. The stability and performance of the model are assessed correctly as well. The ant colony optimization (ACO) method of *Sun et al. (2022)* recommends optimal routes by using travel pattern sequences.

Since the last decade, the techniques of machine learning (ML) are been widely utilized in the realm of tourism. For systematic analysis of tourism data, the support vector regression (SVR) model is used in *Yang & Ren (2024)*. Similarly, the models based on multilayer perceptron (MLP) are used in *Batista e Silva et al. (2018)*, *Li & Law (2020)* for analysis and minimizing the occurrences of errors in tourism-related information. The cutting-edge neural network (NN) was successfully implemented by *Zhang & Tang (2022)* to predict travel routes. The convolutional neural networks (CNN) based approach of *Logesh et al. (2020)* recommends attractions after analyzing the sentiment of the tourists. The model needs the necessary dataset to enhance the effectiveness of the recommendation system. *Cheng (2021)* proposed a commendable travel route commendation system using the distance matching technique (*Raees & Ullah, 2019*). The system considers preference or the user interests to suggest an appropriate destination. The hierarchical multi-clue fusion (HMCF) based system of *Yang et al. (2018)* exploits user-generated content (UGC) for precise prediction. The model improves the accuracy rate in tourism and enhances travel route recommendations. Similarly, a collaborative mining and filtering process (CMFP) based method is proposed in *Nan & Wang (2022)* to reduce the mining cost in travel recommendations. The Bidirectional Encoder Representations from the Transformers (BERT) model of *Wen, Liang & Zhu (2023)* are based on sentiment analysis, whereas the LSTM networks (*Law et al., 2019*) make use of time series forecasting in tourism. The model effectively traces long-term dependencies for demand forecasting in the tourism domain. Though most of the recommendation systems in the literature

focused on route recommendation, there is a pressing need for an effective tourism spot recommendation system based on user's interest, hence this research.

## METHODOLOGY

To assure tourists' satisfaction, the proposed method recommends tourist spots and routes based on a single-token textual input. The method intends to enhance the tourism experience by presenting an optimized route map based on personal preference. The technique utilizes machine learning classifiers to perform each subsequent process effectually. The system's algorithm is shown in Fig. 2, whereas details about each process are presented in the following sub-section.

Preceded by graph representation, a CL is selected while nodes in the region share their information with the CL. The CL keeps a record in a CSV file, which contains each attraction's information. $A_i$ for $i = \{1, 2, ., ., n\}$ of a particular node with spot name SNj; $A_i \in SN_j$. At the CL level, LSTM is trained by the resources/facilities available in the region. Spot name is predicted by feeding the generated word-level sequences of LSTM to the trained SVM. The benefit of using LSTM at CL is that it provides information on related attractions at the nearest tourist spots. Suppose a tourist searches for a facility $f_x$, the LSTM will provide a list of facilities; $[f_x, f_{x+1}, ., ., f_{x+n}]$. The set of attractions or facilities is then forwarded to SVM to predict the spot name for onward operation.

### Grid representation of tourist spots

A tourism map is treated as a graph G, where $G = \{V, E, W\}$ where the nodes V represents the visiting spots, E the edges and W is the weights (distance in Km). A path $p(v1, v2) = [v1, v2, \ldots vn]$ represents the possible permutation between the nodes $v1 \in V$ and $v2 \in V$. By utilizing the Dijkstra's Algorithm, the distance between node $v1$, and $v2$ is calculated as,

$$d(v1, v2) = \min\left(\sum\nolimits_{(x,y) \in T} w(e1, e2)\right) \forall \text{ paths } T \text{ from } v1 \text{ to } v2 \tag{1}$$

where $w(e1, e2)$ is the weight between the edge $e1$ and $e2$. A simple graph illustration of the following list of connected tourism spots is presented in Fig. 3.

[('Spot1', 'Spot4'), ('Spot0', 'Spot4'), ('Spot0', 'Spot2'), ('Spot4', 'Spot3')]

Determining node-to-node weights depends on examining a basic 3 × 3 tourist spot grid represented by nodes. The calculated weights for nodes represent edge connections between nodes in the graph. Thus, node v1 at (0,0) maintains a weight value of five units when connected to node v2 at (0,1), as well as a link weight of three units between node v2 and node v3 at (1,1). We begin the algorithm at node v1 while setting its distance to 0, then compute the minimum routes to neighboring nodes by changing their distance values according to weight information. The system uses Dijkstra's algorithm to discover the most direct path between v1 (0,0) and v9 (2,2) by evaluating several nodes and choosing the shortest connection. This algorithm's shortest distance tracking capability determines an optimal v1-to-v9 path through successive step progression.

---

**The RARE algorithm**

**Input:**
 - G: a 2D Matrix representing the graph
**Variables:**
 - AR: All Routes
 - D: Distances
 - SN: Spot Name
 - V: Vertex (node)
 - E: Edges (D)
**Output:**
 - AR
Input matrix(SN, D)
fun Graph(SN,D):
        G(V,E)←matrix(SN,D)
end
fun clustering (samples xn, clusters k):
        for j=1 to d:
                $C = \frac{1}{n}\sum_{i=1}^{n} x_{ij}$
                $Cltr_i = \ Ed_{min} (\|C, x_i\|)^2$
                $CL = Random(|Cltr_i|_{i=1}^{n})$
        end for
end
fun Info_sharing(Nodes, CL):
        for i in $Cltr_i$:
                $File\_CSV += node_i$
        end for
        CL_CSV=File_CSV
end
Train_Test LSTM_by(CL_CSV['Tourist_Attract_Info'])
Train_Test SVM_(CL_CSV['SN_Strongly_correlated_features'])
AR=DFS(G)

**Figure 2  The rare algorithm.**               

## Cluster leader selection

Detection of clustering patterns using k-means proves effective because it efficiently manages substantial dynamic datasets like those found in tourism data. Unsupervised learning algorithm k-means partitions data into separate clusters based on data similarity distributions. This method's simplicity and computational power suit applications that need rapid and efficient clustering solutions while processing datasets with frequently changing features, like tourist preferences and regional features. Despite the dynamic characteristics of tourism data, k-means analyzes the data consistently into distinct clusters, which help identify key interests, making this method a strong selection for our system. With the k-means clustering approach, a CL node is selected for a region containing k tourist spots. In the process of finding CL, initially k

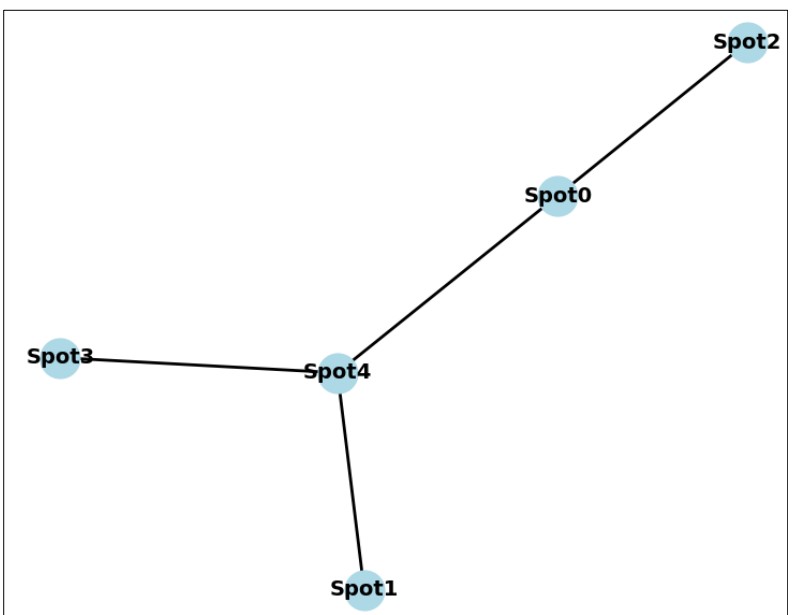

**Figure 3** **A generated graph of five connected nodes.**

centroids-$\{U_1, U_2, \ldots, U_k\}$ are set to a visiting spot $S_x$ which in turn is assigned to the nearest cluster centroid ($C_i$) using the given criteria,

$$C_i = \left\{ S_j : \; E_d \lVert S_j - U_i \rVert^2 \leq E_d \lVert S_j - U_m \rVert^2 \; \forall \; m = 1, \ldots, k \right\} \tag{2}$$

where $E_d$ represents Euclidean distance (*Li et al., 2023*). Next, the mean of the centroid is updated to accommodate the newly added spot information,

$$U_i = \frac{1}{|C_i|} \sum_{Sx \in C_i} S_x \tag{3}$$

where $|C_i|$ represents the number of nodes in the cluster. The convergence is checked to ensure that the centroid has no change or is negligible. For this purpose, convergence is achieved as,

$$\sum_{i=1}^{k} \lVert U_i^{t+1} - U_i^t \rVert^2 < \partial \tag{4}$$

In the given equation, $t$ is the number of iterations and $\partial$ being the threshold value. Our approach used empirical cross-validation testing on the tourism dataset to select the threshold value ($\partial$) for the k-means clustering algorithm. Our study performed cluster analysis under various $\partial$ settings while measuring cluster outcomes to confirm their agreement and suitability for tourist preferences. The tested threshold $\partial$, 0.5, served as the optimal value, generating clusters with high accuracy and efficient computation demands.

It is assumed that all the nodes of a cluster, $n \in Cltr_x$ Have their own attraction and facilities information except the distance measured from the rest of the nodes. Though the

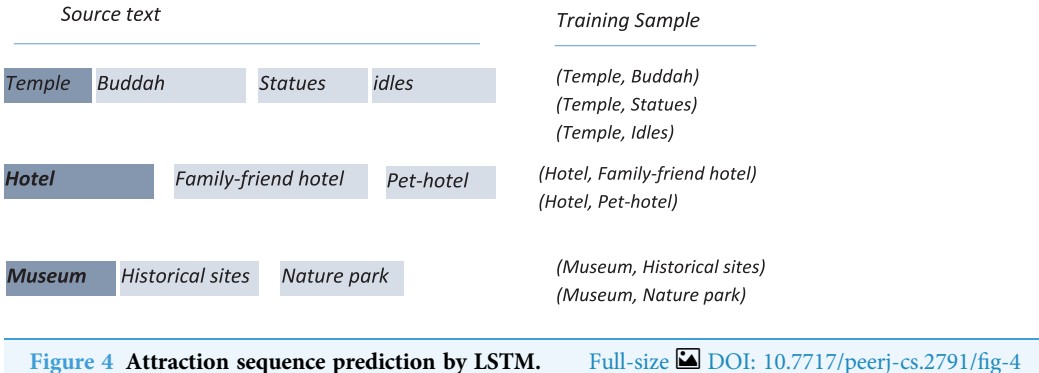

**Figure 4 Attraction sequence prediction by LSTM.**

user may feed explicit distance information, Dijkstra's algorithm may be utilized to find out the distance measures, whereas all the nodes $n \in Cltr_x$ Shared their information with the CL. The dataset $D$ with rows $r$ and columns $C$ at the $CL$ is updated with the new feature $f = [f1, f2, , ,fn]$ and is inserted at the second last column; $C - 2$. For this purpose, $D$ is split into sub-matrices $D_L$ with columns $C_L$ and $D_R$ with $C - C_L$ columns as;

$$D_L = D[:, : C_L] \tag{5}$$

$$D_R = D[:, C_L :] \tag{6}$$

To get the resultant dataset $D'$, $D_L$, the new column $C_f$ and $D_R$ are concatenated as;

$$D' = [D_L, \ C_f, \ D_R] \tag{7}$$

## Tracing relevant facilities by LSTM

In the proposed method, the deep learning-based approach is followed to recommend attraction-related information in the textual entry of a tourist. The method predicts travel preferences after analyzing the input query by utilizing the sequence modeling and attention mechanisms. LSTM is particularly exploited to suggest optimized tourist attraction facilities based on fashion preferences. At the CL level, LSTM is trained based on the attraction information offered by region nodes. Taking the input token from the tourist, LSTM presents related attraction information of the tourist spots, as shown in Fig. 4. The output of LSTM is further fed to SVM to name the appropriate tourist spot.

LSTM is the enhanced recurrent neural network (RNN) deep learning model where the vanishing gradient issue is overcome by backpropagation (*Karmiani et al., 2019*), where RNN is an artificial neural network (ANN). In RNN, the nodes are connected to form a temporal sequence directed graph. At each stage, the output of RNN is to coincide to generate new output. However, the network suffers from the issue of preserving the extensive past information (*Supakar, Satvaya & Chakrabarti, 2022*). LSTM is the modified and advanced form of RNN (*Supakar, Satvaya & Chakrabarti, 2022*) with the ability to

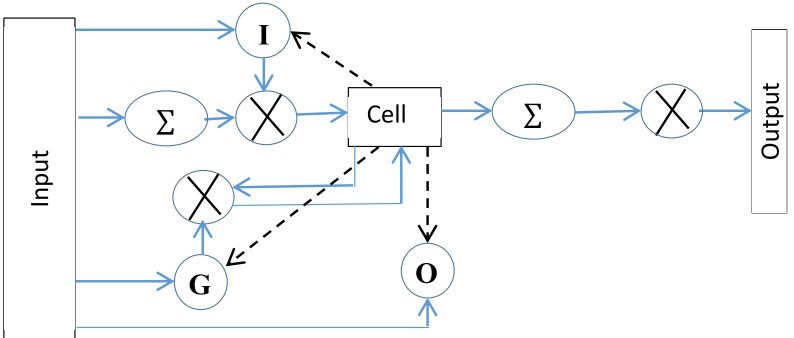

**Figure 5 The standard LSTM structure, represents activation function.**

back-propagate to retain a large number of related information If" is the hidden node at a particular time instance t, its output Oh t is given as

$$O_h^t = a^h \left( \sum_{i=0}^{I-1} CW^{hi} x_t^i + \sum_{m=0}^{Th-1} u^{hm} b_{t-1}^m + c^h \right) \tag{8}$$

where, $a^h$ is the activation function at node h and $c^h$ is the bias value at a particular node". $I$ and $Th$ represent the total number of inputs and the number of nodes in a layer, respectively. $x_t^i \in R$ is the ith input occurring at node $h$ at time $t$, $CW \in R$ connection weight between input 'i' and node 'h'. Similarly, $u^{hm} \in R$ is the connection weight between the nodes h and m, which belong to the same layer, whereas $b_{t-1}^m$ is the output of at m node in time t − 1.

A simple architecture of LSTM is represented by three gates with a memory cell. The gates control information to and from the cell. The input gate manages the input of the flow of information into the memory. The forget gate controls how long information be retained in the cell, whereas the output gate is responsible for generating the output of the LSTM. The working of LSTM, as depicted in Fig. 5 is represented mathematically as follows,

$$I_t \;=\; \sigma(W_i[h_{t-1}, x_t] + b_i) \tag{9}$$

$$F_t \;=\; \sigma(W_f[h_{t-1}, x_t] + b_f) \tag{10}$$

$$O_t \;=\; \sigma(W_O[h_{t-1}, x_t] + b_O) \tag{11}$$

$$Ct = Af \,^\circ Ct - 1 + Ai \,^\circ \sigma_c (Wcxt + Ucht - 1 + bc). \tag{12}$$

In the given equations, input, forget, and output gates are represented by I, F and O, respectively. Weight matrices are represented by W (*e.g.*, $W_i$, $W_f$, $W_O$ are weight matrices for input, forget and output neuron, respectively). The previous state of LSTM, at timestamp t − 1, is represented by $h_{t-1}$ whereas the bias value for a gate is represented by

$b_x$. The network's output activation functions (softmax) is represented by $\sigma$. Similarly, $\sigma_c$ is the cell's hyperbolic tangent activation function. $Wc$ and $Uc$ is the matrix weights of the connections and recurrent connections respectively to the memory cell of LSTM. Af, Ai and Ct are the activation functions of the memory cell's forget gate, input gate and state vector. b represents bias it may be bi, bf, bo, or bc which shows bias at input, forget, output gate, or at the memory cell, respectively. The operation (°) represents the Hadamard product.

## Tracing of tourist Spot by SVM

The SVM classifier is one of the effective supervised ML models used for regression and classification. In its simple form (binary classification), the model intends to find a hyperplane between two classes with class labels $L_i \in \{-1, 1\}$ for the given training dataset $\{x_i, L_i\}_{i=1}^n$ where $x_i \in \mathbb{R}^d$ the feature vector. The hyperplane is given as,

$$w.x + b = 0$$

where $w$ is the weight and $b$ the bias.

For an unknown input point $x_x$, the decision function f($x_x$) is given as,

$$f(x_x) = sign(w.x_x + b) \tag{13}$$

In case of high dimensional data, the decision function be as,

$$f(x_x) = sign\left(\sum_{i=1}^n \phi_i c_i (x_i.x_x) + b\right) \tag{14}$$

where $\phi_i$ is the Lagrange multiplier.

In the proposed method, the SVM is trained by features strongly correlated with the spot name. The attraction facilities of tourist spots- $X = \{f1, f2, \ldots fn\}$ are taken as features, whereas the names of spots as labels; $N = \{c1, c2, c3, ., cn\}$. The features, as found out by LSTM, are fed to the SVM to prompt the most appropriate tourist spot name, $SN$. Multiple elements determine the LSTM algorithm's decreasing accuracy during extended sequence length prediction. The algorithm's nature makes it difficult to process elaborate sequences because the vanishing gradient problem affects it even with its built-in long-range dependency capabilities. Longer sequences cause the model to lose its ability to capture important sequence data from previous parts, making predictions less precise. The escalating number of sequence configurations leads to improved model overfitting, which degrades the model's ability to adapt to new data.

## Routes recommendation by DFS

The effective DFS is exploited for route recommendation (*Hagerup, 2020*; *Gers & Schmidhuber, 2001*; *Gedela & Karthikeyan, 2022*; *Ni et al., 2021*; *Li et al., 2021*; *Qiao et al., 2024b*; *Zeng et al., 2015*; *Paulson & Tzanavari, 2003*; *Abadi et al., 2016*; *Rafay, Suleman & Alim, 2020*). The algorithm offers a comprehensive exploration of all possible routes based on the priorities and constraints of a user. Once a spot name is suggested, routes from the predicted spot to the rest of the spots in a *Cltr* are searched out by DFS.

**The DFS algorithm**

**Input:**
- G: a 2D Matrix representing the graph

**Variables:**
- P: the current Path
- AP: All Paths
- Vis: Boolean variable to show if a node is visited or not
- $v_i$: Initial vertex (node)
- $v_c$: Current vertex (node)

**Output:**
- AP

**Begin**
  Vis($v_i$)= False ∀ $v_i$ ∈ V
  P=[]
  AP=[[]]
  function DFS ($v_i$):
     Vis($v_i$)=True
     P← $v_i$
     If $v_i$ == $v_c$ then
       AP←P
     else
       For each neighbor $v_c$ in G:
         If not Vist($v_c$):
           DFS($v_c$)
       End for
       Vis($v_c$)=False
       P.pop()
  return (AP)
**End**

**Figure 6  The routes finding DFS algorithm.**

    The algorithm is designed to explore all possible paths recursively till the end of the destination node. Let G = {V,E} represents the tourist spots in a *Cltr* with nodes $V = \{v1, v2, ., ., vn\}$ and edges $E = \{e1, e2, ., ., en\}$. Any edge $e_i \in E$ is a pair (v1, v2), where $v1, v2 \in V$. The steps of the algorithm are presented in Fig. 6. Suppose a cluster $Cltr_x$ Has five such nodes (A, B, C, D, E) containing the required or matching tourism facilities; as predicted by the LSTM, the possible routes, as recommended by DFS, are shown in Fig. 7.

    In a nutshell, key contributions of this research work include: (1) a machine learning framework is proposed to provide tourist spots with travel routes based on the requirements or priorities of the tourists. (2) The method is implemented in a case-study project by using a real-world dataset and augmented with the required attributes. (3) The results obtained are analyzed, and they show promising improvements in contemporary systems.

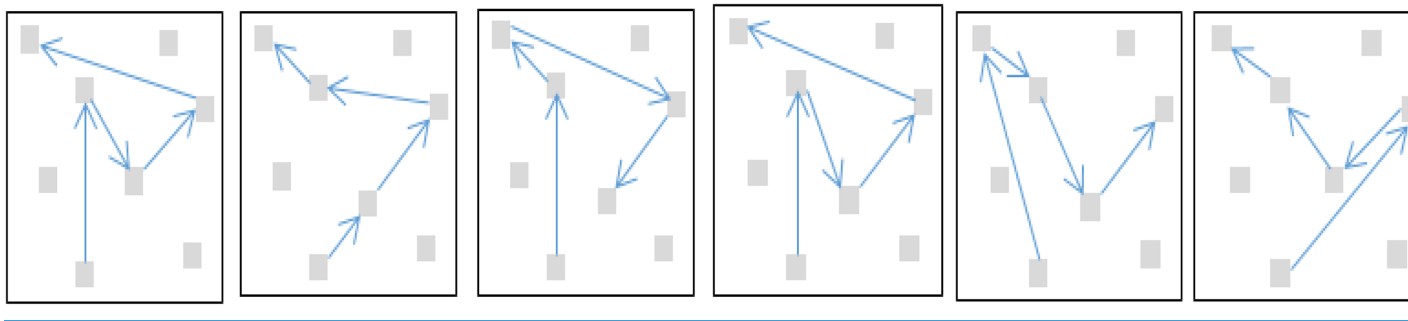

**Figure 7 The recommended routes as generated by the DFS algorithm.**    

## IMPLEMENTATION DETAILS

The proposed method is implemented in VSCode using various advanced machine learning libraries, including TensorFlow, Keras, and Python networks. The system contains four modules: representation, selection, information tracing and spot-name tracing. In the *representation* module, data is taken from the dataset and is represented in 2D graph form using Dijikstra's algorithm from the Networkx library (*Chollet, 2015*; *Jabbar, Abass & Hasan, 2023*). In the said module, the spot names and their distances are taken as inputs. A simple demonstration of the module is shown in Fig. 8. The computational complexity of the algorithms varies. LSTM has higher complexity due to its recurrent nature, while SVM can be slow with large datasets. DFS is efficient for small datasets but could be optimized for larger ones.

In the *selection* module, the CL is selected out of 'n' nodes in the cluster using the centroid-based approach. Each node 'n' shares its available attraction information with the CL, which is appended to the dataset and is stored in a CSV file. Some of the tourism information of a CL in tabular form is presented in Table 1.

In the *information-tracing* module, LSTM is trained by the available information $I_i$ of the node $N_i$ where $(I_i, N_i) \in CL$. LSTM is an effective model for learning context-free and context sensitive languages (*Gers & Schmidhuber, 2001*). LSTM retains previously learned knowledge through a memory cell at the hidden layer. In the proposed system, LSTM is used to map input sequence $I\beta_i = (s_1, s_2 \ldots s_n)$ to output sequence $\Omega = (os_1, os_2, \ldots .os_n)$. By applying unit activation iteratively, where $I\beta$ and $\Omega$ represent tourist information by which it is trained. A predicted sequence $\Omega_x$ is concatenated to the formerly predicted sequence $\Omega_{x-1}$ to generate the next sequence $\Omega_{x+1}$. This sequence prediction process is repeated till the last token term; $k_1$ as shown in the following equations for the predicted sequence.

$$PS_r = \sum_{i=0}^{k1} \omega(\Omega i). \tag{15}$$

An illustration of generated *PS* is shown in Fig. 9 by entering Restaurant as an input to the LSTM.
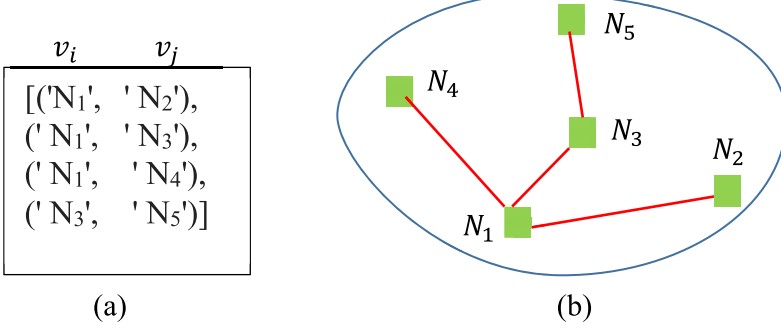

|   $v_i$   |   $v_j$   |
|-----------|-----------|
| [('N₁', | 'N₂'), |

$$[(\text{'N}_1\text{'}, \ \text{'N}_2\text{'}),$$
$$(\text{'N}_1\text{'}, \ \text{'N}_3\text{'}),$$
$$(\text{'N}_1\text{'}, \ \text{'N}_4\text{'}),$$
$$(\text{'N}_3\text{'}, \ \text{'N}_5\text{'})]$$

(a)  (b)

**Figure 8** **(A) List of the connected nodes and (B) graph representation of the algorithm.**

**Table 1 Tourism information maintained at CL level.**

| Name | State | Type | Popularity | Best time to visit | Currency | Culture | Distance (Km) |
|------|-------|------|------------|--------------------|----------|---------|----------------|
| Taj Mahal | Uttar Pradesh | Historical | 8.6919062031 | Nov–Feb | Indian (Rs) | Indian | 123 |

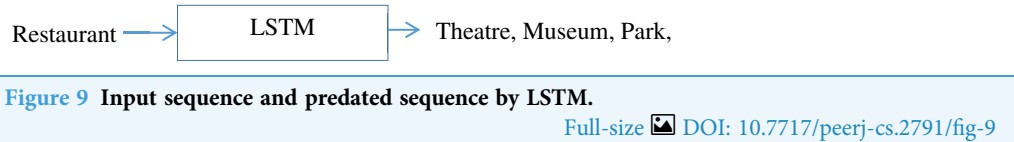

**Figure 9** **Input sequence and predated sequence by LSTM.**

The spot-name tracing module deals with the tracing of spot names based on the generated *PS* of LSTM. To predict the optimal Spot-Name, the features train SN, SVM $x_i \in X$, $X$ being the training dataset and $y_i \in Y$ as class label where $Y = X['SN']$.

## EXPERIMENTATION AND RESULT ANALYSIS

Unlike other deep learning methods where a classifier is trained, validated and tested by different dataset splits, the proposed system is trained by all tuples of the dataset that lies at a CL. The entire dataset was used solely to ensure the most optimal results. A data augmentation approach was implemented to boost model robustness and enhance generalization because we added artificial dataset elements to the input data. Our approach expanded rows through multiple data transformation tools, including random dimensional changes, scaling operations and flipping mechanics on tourist location records. We developed synthetic features (columns) from historical travel data to analyze potential tourist behavior characteristics, including seasonality choices and travel window behaviors. Observational tests verified the usefulness of the enlarged dataset, which resulted in a 15% increase in model accuracy, thus validating the performance-enhancing effects of these data improvements. The original Kaggle dataset was augmented heuristically by 83 rows and 12 columns. The system was evaluated 52 times by entering textual tourism query—$TQ_i$ of variant dimensions; $TQ_i = \{1, 2, ., ., 7\}$. A $TQ_x$ was

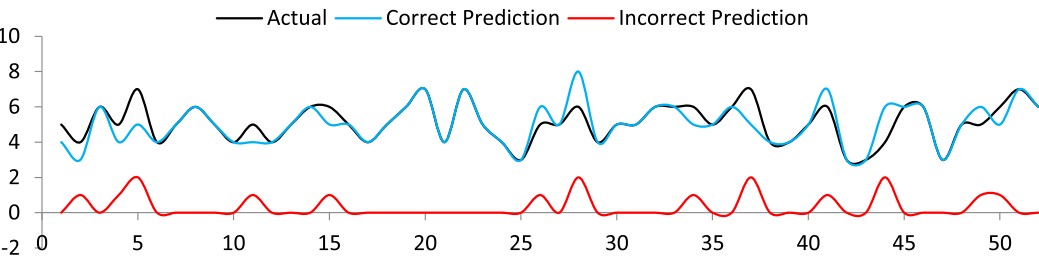

**Figure 10  Actual sequence, correct and incorrect sequence prediction by LSTM.**

**Table 2  Results of the sequence prediction by LSTM.**

| Accuracy (%) | Precision | F1 score | Specificity | Sensitivity |
|---|---|---|---|---|
| 82 | 0.9 | 0.89 | 0.5 | 0.88 |

obtained by feeding a random attraction term to the information tracing module. Though each of the evaluation iterations assessed the entire system, from sequence-related tourism information to possible routes, the results of each section were analyzed individually. The 52 queries used for evaluation were carefully selected to cover a broad spectrum of tourism scenarios, ensuring that they reflect diverse tourist preferences and behaviors. These queries were designed to represent various trip types, including cultural, historical, and recreational tourism, and they included different levels of complexity, such as simple requests for specific attractions or more detailed queries involving routes between multiple spots. The selection process aimed to ensure the model was tested under conditions that closely mirror real-world tourism planning, capturing the full range of user preferences and geographic contexts.

LSTM was assessed to generate the available tourist attraction information in 1 and 7 sequences. A user only needs to input a textual term and the number of sequences to be generated. The accuracy of LSTM was up to the mark for a lesser number of sequences. However, irrelevant terms were predicted in case of a greater number of sequences. This is clear from the correct and irrelevant sequence prediction graph shown in Fig. 10. As a whole, reasonable accuracy, precision, and F1 score were obtained, as shown in Table 2. The formulas for precision, sensitivity, specificity and F1 score (*Gedela & Karthikeyan, 2022*) are follows.

$$\text{Precision} = \frac{TP}{TP + FP} \qquad (16)$$

$$\text{Sensitivity} = \frac{TP}{TP + FN} \qquad (17)$$

$$\text{Specificity} = \frac{TN}{TN + FP} \qquad (18)$$

$$\text{F1 score} = 2 \times \frac{Precision \times Sensitivity}{Precision + Sensitivity}. \qquad (19)$$

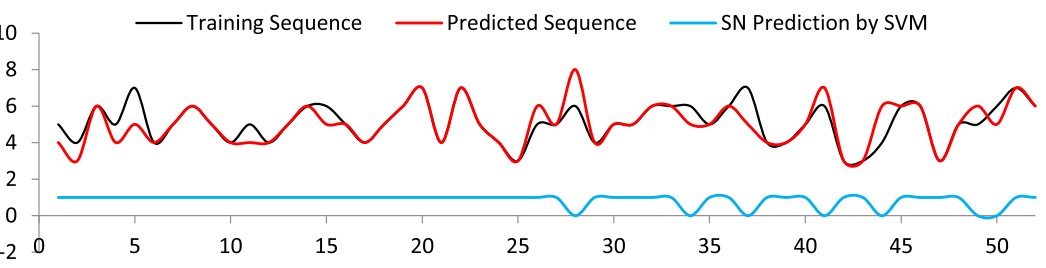

**Figure 11  Spot name prediction with respect to training sequence, and predicted sequence.**

**Table 3  Results of the SVM model.**

| Accuracy (%) | Precision | F1 score | Specificity | Sensitivity |
|---|---|---|---|---|
| 90 | 0.93 | 0.94 | 0.4 | 0.95 |

**Table 4  Details of the state-of-the-art recommendation methods.**

| Author(s) | Title | Method used | Remarks |
|---|---|---|---|
| *Ni et al. (2021)* | "Collaborative Filtering Recommendation Algorithm Based on TF-IDF and User Characteristics" | Collaborative filtering based on (TF-IDF) and fuzzy membership | Needs to find similar users and resources |
| *Li et al. (2021)* | "Intelligent recommendation model of tourist places based on collaborative filtering and user preferences" | Hamming and Jeffries-Matusita distance approach | User preferences is supported and the issue of sparse data is addressed |
| *Qiao et al. (2024a)* | "Tourist Recommender System using Hybrid Filtering" | Multi-criteria based containing content, C-filtering, and Ontology | Predicts the ratings of demography and POIs to satisfy needs of users |
| *Zeng et al. (2015)* | "Optimal Route Search with the Coverage of Users' Preferences" | Algorithm based on A* and heuristic function | The weighted user preference is considered in route researching, proposes convergence solution for optimal route. |
| *Paulson & Tzanavari (2003)* | "Combining Collaborative and Content-Based Filtering Using Conceptual Graphs" | Conceptual graphs and dataset contents for prediction and error reduction | Support only a limited fuzzy value fields and tourism attractions |
| RARE | The proposed system | Multi-model based method including SVM, LSTM and DFS | Tourist preference is supported to trace spot and routes. |

The generated sequences are forwarded to SVM to predict the appropriate spot names. Most of the spot name classes were accurately predicted. In the odd cases where the predicted sequences are fewer than the training sequences, incorrect class names were expected, see Fig. 11. Analysis of the results obtained for the SVM is presented in Table 3.

The DFS algorithm precisely depicts all the routes between a current node, $n_c$ and the predicted destination route $n_p$. Hence, the accuracy of the algorithm was simply 100%. Our augmentation technique that builds synthetic features using historical information has the potential to strengthen existing patterns within the original dataset. Specific tourist

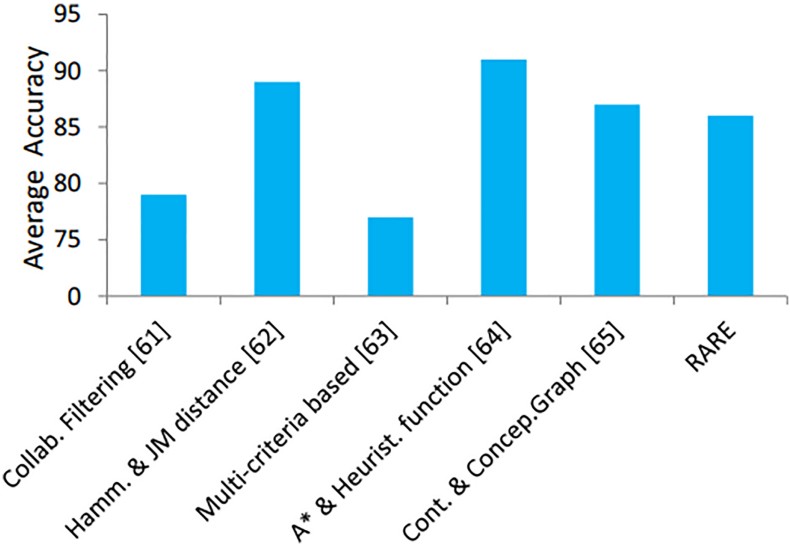

**Figure 12 Accuracy comparison of the state-of-the-art recommendation methods.**

demographic biases often become more pronounced when we use historical data to determine popular destinations. Instrumented modifications to original data through special transformations such as scaling and random rotations might not adequately interpret natural modifications in tourism behavior patterns. The model's generalization capability faces challenges when using the data for areas lacking adequate representation within the initial dataset. Real-world data performance assessments led to continual updates of the augmentation system that reduced the effect of biases discovered in the process.

In addition, the proposed method is compared with some up-to-date, cutting-edge approaches, see Table 4. Comparison in terms of accuracy is shown in Fig. 12. The performance of RARE surpasses contemporary methods presented in Table 4 because it utilizes SVM and LSTM together with DFS in its architectural design. RARE defeats traditional techniques such as collaborative filtering and hybrid filtering by implementing LSTM to track unique visitor preferences. The DFS algorithm delivers efficient route optimization through tourist priority interests and geographic limitations, which other methods like A* do not handle properly. The unified framework of models enables RARE to generate travel guidance that surpasses the accuracy, adaptability, and tailored nature of standalone approaches.

The model does not currently incorporate temporal data, such as opening and closing times, which could affect the accuracy of recommendations for time-sensitive travel plans. We plan to address this limitation in future work by integrating such data to improve the model's relevance in dynamic tourism scenarios. Future work will also focus on integrating GIS data to enable the automatic detection and localization of tourist spots, reducing reliance on explicit user input and further improving the system's usability.

## CONCLUSION

Tourism is the modern-day valuable entertainment. Tourism adventure can be made more attractive if routes and spots are based on personal preference. This research proposes a multimodal approach for tourist routes and spot recommendations based on the personal priority of the visitor. The method exploits the approach of a right algorithm for the right task. To keep the attraction information of all the nodes, the k-mean clustering algorithm is used to select a cluster leader in a particular zone. The LSTM model is trained to generate relevant attraction information, whereas the SVM classifier is used to suggest the most optimal spot name based on the generated input sequences. By inputting a simple textual demand term, the matching and relevant spots are suggested, as well as presenting travel routes to the various tourist spots. The method is implemented in a Python based project and is evaluated by an augmented real-world dataset. The proposed method is systematically assessed and compared with the contemporary state-of-the-art recommendation systems. The satisfactory accuracy revealed that the process is applicable for enhancing tourism. The system has strong potential for real-world applications in large-scale tourism datasets, such as in smart tourism platforms. By leveraging its ability to process diverse tourist preferences and optimize routes, it can scale to accommodate millions of users. Future improvements, including better data integration and performance optimization, will enhance its scalability for large, dynamic datasets.

Though the research presents a novel approach for tracing tourist spot based on user's preferences, the system fails to consider the opening and closing times, often critical for tourism planning. As our future strategy, we are scheming to enhance the model. Moreover, the GIS data will be incorporated into the dataset to automatically locate tourist spots without explicit inputting. As a whole, the research contributes to the realm of tourism and paves the way for future research.

### Funding

This study was supported by the Annual project of Philosophy and Social Science Planning of Henan Province in 2024 (No. 2024BJJ166). The funders had no role in study design, data collection and analysis, decision to publish, or preparation of the manuscript.

### Grant Disclosures

The following grant information was disclosed by the authors:
Annual Project of Philosophy and Social Science Planning of Henan Province in 2024: 2024BJJ166.

### Competing Interests

The authors declare that they have no competing interests

## Author Contributions

- Ling Luo conceived and designed the experiments, performed the experiments, analyzed the data, performed the computation work, prepared figures and/or tables, authored or reviewed drafts of the article, and approved the final draft.

## Data Availability

The code and raw data are available in the Supplemental Files and at GitHub and Zenodo:

- https://github.com/halowisata/ViVe-Machine-Learning-Flask.
- Ling, L. (2025). RARE: Right Algorithm for the Right Errand; A multi-model machine learning-based approach for tourism routes and spots recommendation [Data set]. Zenodo. https://doi.org/10.5281/zenodo.15038384.

## Supplemental Information

Supplemental information for this article can be found online at http://dx.doi.org/10.7717/peerj-cs.2791#supplemental-information.

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
