# Peer review of "RARE: right algorithm for the right errand; a multi-model machine learning-based approach for tourism routes and spots recommendation"

_PeerJ Computer Science, doi:10.7717/peerj-cs.2791_

## Round 0.1 · original submission · Major Revisions

Dear authors,


Reviewers have now commented on your article. We do encourage you to address the concerns and criticisms of the reviewers with respect to reporting, experimental design, and validity of the findings and resubmit your article once you have updated it accordingly. Following should also be addressed:

1. Please write research gap and the motivation of the study. Evaluate how your study is different from others. Please highlight the originality, novelty, and advantages of the proposed and used methods. More recent literature should be examined. The works presented in Introduction section seem old.
2. Some paragraphs are too long tore read. They should be divided into two or more for cpmprehensibility and readability.
3. Many of the equations are part of the related sentences. Attention is needed for correct sentence formation.
4. Equations should be used with correct equation number. Please do not use “as follows”, “given as”, etc. Explanation of the equations should also be checked. All variables should be written in italic as in the equations. Their definitions and boundaries should be defined. Necessary references should be provided.
5. All of the values for the parameters of all algorithms should be given.
6. All references should be written according to journal referencing style. See for example: [45].
7. Please pay special attention to the usage of abbreviations. Spell out the full term at its first mention, indicate its abbreviation in parenthesis and use the abbreviation from then on.
8. Please provide information about implementation details regarding both software and hardware.

Best wishes,

Reviewer 1 ·

Basic reporting

The manuscript "RARE: Right Algorithm for the Right Errand; A Multi-Model Machine Learning-Based Approach for Tourism Routes and Spots Recommendation" is an innovative contribution to tourism recommendation systems. It is commendable that machine learning algorithms such as LSTM, SVM, and DFS are integrated to create a personalized and optimized travel experience. The comprehensive approach, from data representation to route optimization, demonstrates a deep understanding of the challenges in modern tourism and presents a promising solution. There are some concerns made by the reviewers and addressing these concerns can improve the manuscript's clarity, depth, and impact, making it a significant contribution to tourism recommendation systems research.

Experimental design

1. The Abstract could be improved by rephrasing "Tourism attractions and spot recommendations are becoming the demand of the day besides optimal route suggestions" to make it more concise and impactful.
2. To enhance its appeal, the abstract should more explicitly highlight the novelty of integrating multiple machine learning algorithms for personalized tourism recommendations.
3. In the methodology, the sentence "distance between node v1 and v2 is found" should be revised to "distance between node v1 and v2 is calculated" to ensure grammatical accuracy.
4. The k-means clustering approach requires additional explanation to justify its selection over alternative clustering methods, especially for datasets with dynamic features.

Validity of the findings

5. The Grid Representation section could benefit from a brief example demonstrating how weights (distances) between nodes are computed using Dijkstra’s algorithm.
6. Adding citations for the machine learning libraries (e.g., TensorFlow, Keras) and the DFS algorithm would strengthen the credibility of the Implementation Details section.
7. Figures 2, 3, and 8 need consistent labeling, and their captions should succinctly describe their significance in the methodology.
8. Figure 7 would benefit from a detailed caption explaining how DFS generates routes, enhancing reader comprehension.
9. The choice of hyperparameters in k-means clustering, such as the threshold value (∂), should be supported with theoretical or empirical justifications.

Additional comments

10. The dataset augmentation process should be elaborated to detail how the additional rows and columns were created and their impact on model performance.
11. Potential biases introduced during dataset augmentation should be acknowledged and discussed in the manuscript.

Reviewer 2 ·

Basic reporting

.

Experimental design

.

Validity of the findings

.

Additional comments

1. In the Introduction section, replacing "tourists used to exploit guidebooks" with "tourists traditionally relied on guidebooks" conveys a more formal tone and improves readability.
2. The phrase "the map of tourist spots is deemed as a 2D grid" could be revised to "the map of tourist spots is represented as a 2D grid" for better clarity and alignment with technical terminology.
3. The Results section requires a deeper analysis of why LSTM’s accuracy declines for longer sequence predictions and suggestions for mitigating this issue.
4. Including confidence intervals or standard deviations for accuracy, precision, and F1-score metrics in Tables 2 and 3 would provide a more robust model evaluation.
5. The factors contributing to the proposed model’s superior performance compared to state-of-the-art methods in Table 4 should be briefly discussed.
6. Clarify how the 52 queries used for evaluation were selected, ensuring they reflect diverse and realistic tourism scenarios.
7. The exclusion of temporal data (e.g., opening and closing times) is a limitation that should be addressed or acknowledged in the discussion.
8. Future work could expand on integrating GIS data, significantly enhancing the system’s ability to locate tourist spots automatically without explicit user input.
9. Discussing potential real-world applications and the system's scalability for large-scale tourism datasets would add valuable context.
10. A grammatical issue in the Conclusion section: "the system fails to consider the opening and closing time often required in planning tourism" should be revised to "the system fails to consider the opening and closing times, which are often critical for tourism planning."
11. The Conclusion section could provide actionable insights, such as strategies for enhancing the system with temporal and real-time data, to make the study’s contributions more tangible.
12. The Implementation Details section needs an additional explanation of the computational complexity of the proposed algorithms, especially their scalability in large-scale datasets.

---

## Round 0.2 · accepted · Accept

Dear Authors,

Thank you for clearly addressing the reviewers' comments. Your manuscript now seems sufficiently improved and ready for publication.

Best wishes,

Reviewer 1 ·

Basic reporting

no comment

Experimental design

no comment

Validity of the findings

no comment

Reviewer 2 ·

Basic reporting

.

Experimental design

.

Validity of the findings

.

Additional comments

.